# Characterization of Volatile Profiles and Correlated Contributing Compounds in Pan-Fried Steaks from Different Chinese Yellow Cattle Breeds through GC-Q-Orbitrap, E-Nose, and Sensory Evaluation

**DOI:** 10.3390/molecules27113593

**Published:** 2022-06-02

**Authors:** Meng Wei, Xiaochang Liu, Peng Xie, Yuanhua Lei, Haojie Yu, Aiyun Han, Libin Xie, Hongliang Jia, Shaohua Lin, Yueyu Bai, Baozhong Sun, Songshan Zhang

**Affiliations:** 1Institute of Animal Sciences, Chinese Academy of Agricultural Sciences, Beijing 100193, China; weimeng@caas.cn (M.W.); lxc_cau@163.com (X.L.); seulbird@163.com (P.X.); leiyuanhua-1@163.com (Y.L.); haojie-yu@foxmail.com (H.Y.); baozhongsun@163.com (B.S.); 2Chemical Engineering Institute, Shijiazhuang University, Shijiazhuang 050035, China; irene0001@126.com (A.H.); xielibin827@163.com (L.X.); 3Department of Food and Biological Engineering, Beijing Vocational College of Agriculture, Beijing 102442, China; jiahongliang@bvca.edu.cn (H.J.); lsh_hp@sina.com (S.L.); 4Henan Animal Health Supervision, Zhengzhou 450046, China; baiyueyu666@sina.com; 5School of Agricultural Sciences, Zhengzhou University, Zhengzhou 450001, China

**Keywords:** Chinese yellow cattle, volatile organic compounds, GC-Q-Orbitrap, E-nose, sensory evaluation, multivariate statistical analysis

## Abstract

This study focused on characterizing the volatile profiles and contributing compounds in pan-fried steaks from different Chinese yellow cattle breeds. The volatile organic compounds (VOCs) of six Chinese yellow cattle breeds (bohai, jiaxian, yiling, wenshan, xinjiang, and pingliang) were analyzed by GC-Q-Orbitrap spectrometry and electronic nose (E-nose). Multivariate statistical analysis was performed to identify the differences in VOCs profiles among breeds. The relationship between odor-active volatiles and sensory evaluation was analyzed by partial least square regression (PLSR) to identify contributing volatiles in pan-fried steaks of Chinese yellow cattle. The results showed that samples were divided into two groups, and 18 VOCs were selected as potential markers for the differentiation of the two groups by GC-Q-Orbitrap combined multivariate statistical analysis. YL and WS were in one group comprising mainly aliphatic compounds, while the rest were in the other group with more cyclic compounds. Steaks from different breeds were better differentiated by GC-Q-Orbitrap in combination with chemometrics than by E-nose. Six highly predictive compounds were selected, including 3-methyl-butanal, benzeneacetaldehyde, 2-ethyl-6-methyl-pyrazine, 2-acetylpyrrole, 2-acetylthiazole, and 2-acetyl-2-thiazoline. Sensory recombination difference and preference testing revealed that the addition of highly predictive compounds induced a perceptible difference to panelists. This study provides valuable data to characterize and discriminate the flavor profiles in pan-fried steaks of Chinese yellow cattle.

## 1. Introduction

China has been breeding yellow cattle for a long time and exhibits diverse yellow cattle resources. According to the Chinese Catalogue for Livestock and Poultry Genetic Resources [1], there are 55 indigenous yellow cattle breeds and ten improved breeds in China. Traditionally, yellow cattle in China were used as draft animals, but with the development of mechanization, some elite breeds with high meat yield and quality have been bred for meat production. Beef from these breeds is appreciated by Chinese consumers for its tenderness and desirable flavor. Previous studies revealed that beef flavor is an essential factor that determines consumers’ acceptability [2] and relates to cattle breeds. The flavor attributes of worldwide cattle breeds (e.g., Angus, Holstein, and Simmental), and some local breeds (e.g., Wagyu and Hanwoo) have been well studied [3,4,5,6]. However, the flavor of Chinese yellow cattle has been scarcely studied. The flavor varieties among breeds lead to different consumer preferences and market prices. Therefore, it is necessary to characterize the flavor profiles and correlated contributing compounds of Chinese yellow cattle to clarify flavor differences among breeds, which can further lay a foundation for the establishment of a fair market and the optimization of breeds.

Beef flavor is mainly associated with numerous volatile organic compounds (VOCs). More than 800 beef VOCs have been identified [7], including aldehydes, alcohols, ketones, acids, esters, etc. These compounds are derived from a series of chemical reactions during heating, including the Maillard reaction, amino acid degradation, lipid oxidation, and interactions of their intermediates [8]. Among the numerous volatiles, the odor-active compounds play roles in beef flavor. Sohail et al. (2022) summarized 78 and 90 odor-active volatiles in cooked beef from the Maillard reaction and lipid degradation, respectively [9]. It is worth investigating how these active substances affect sensory evaluation. In recent research, the correlation between VOCs and sensory attribute has been analyzed by multivariate statistical methods to find compounds highly predictive of sensory attribute. Zhang et al. (2018) correlated eight volatiles to the flavor of chicken broth [10]. Dubrow et al. (2022) revealed that ten volatiles in strawberry preserves significantly impacted overall acceptability [11]. Such correlation analyses provide preliminary information on identifying potential flavor contributors.

VOCs in beef are commonly analyzed by gas chromatography (GC) coupled with triple quadrupole mass spectrometer (MS), but this approach is limited by low resolution [12]. In order to capture the exact information of VOCs in beef, high-throughput screening mass spectrometry is needed. GC-Q-Orbitrap spectrometry is a novel technology that provides a high resolving power and mass accuracy [13]. GC-Q-Orbitrap has been proved to be more sensitive than GC-triple-quadrupole in identifying traces of pesticide residues and contaminants in food [14]. Liu et al. (2022) used GC-Q-Orbitrap to explore the aroma compounds in roasted mutton [15]. In addition, the electronic nose (E-nose) is a non-destructive testing technology used to detect and identify complex volatiles with a gas sensor array and pattern recognition system. It is effective, easy to implement, and not computationally expensive [16]. E-nose and GC-MS have been applied to analyze the aroma compounds of Chinese local chicken [17]. Therefore, the primary goal of this study was to apply GC-Q-Orbitrap and E-nose to identify the flavor profiles from different Chinese yellow cattle breeds and establish associations with the sensory evaluation to find contributing flavor compounds.

In this study, strip loin steaks from six yellow cattle breeds that are commonly available on the market were selected. Volatile profiles of pan-fried steaks from these cattle breeds were determined by GC-Q-Orbitrap and E-nose combined with multivariate data analysis. Partial least squares regression (PLSR) analysis between VOCs and sensory evaluation was carried out to obtain volatiles highly predictive of the flavor of Chinese yellow cattle. Sensory recombination difference and preference testing were performed to verify whether these highly predictive compounds contributed significantly to the beef flavor.

## 2. Results and Discussion

### 2.1. Volatile Profiles of Pan-Fried Steaks Obtained by GC-Q-Orbitrap

Qualitative and quantitative analyses of VOCs in pan-fried steaks from different yellow cattle breeds are listed in Appendix A. A total of 92 VOCs were identified by GC-Q-Orbitrap and are shown in the heatmap (Figure 1a). The VOCs identified in the present study were much more diverse than those in previous studies, which detected 24 and 65 VOCs in beef using GC-MS, respectively [18,19]. This was attributed to the better peak resolution and higher resolution of GC-Q-Orbitrap compared to GC-MS. The identified compounds were classified into ten groups, including twenty-two aldehydes, twenty-one hydrocarbons, twelve N-containing heterocyclic compounds, eight S-containing compounds, seven alcohols, six ketones, six O-containing heterocyclic compounds, three fatty acids, two esters, and five other substances. As shown in Figure 1a, the heat map described the flavor profile of each breed. The color intensity, which was based on a normalized scale from a maximum of 4 (red color) to a minimum of −4 (blue color), represented the level of contents. YL and WS had higher contents of most alcohols, aldehydes, and high-molecular-weight hydrocarbons. BH, JX, XJ, and PL contained more pyrazines, low-molecular-weight hydrocarbons, branched aldehydes, S-containing compounds, and O-containing heterocyclic compounds. Figure 1b shows the concentration ratios of each group. The most abundant group in all steaks was aldehydes (31.8–60.7%), followed by hydrocarbons (11.5–37.1%), fatty acids (0.1–17.0%), N-containing heterocyclic compounds (4.5–12.0%), ketones (3.9–8.7%), alcohols (1.4–8.3%), S-containing compounds (0.8–3.0%), O-containing heterocyclic compounds (0.6–1.1%), and esters (0.006–0.4%). 

Aldehydes represented a large group of VOCs found in the pan-fried steaks. As shown in Appendix A and Figure 1a, WS was revealed to be the most abundant in straight-chain aldehydes. Straight-chain aldehydes are the major degradation products of unsaturated fatty acids [20]. Some of them are indicators of fatty acid oxidation, contributing unpleasant flavors at very high levels [21]. Hexanal (2.6–12.5 μg/g) was the predominant straight-chain aldehyde. The result was consistent with previous studies, which reported that hexanal was the major compound in cooked meat [22,23]. This is probably due to its multiple origins, either from the oxidation of oleic acid or linoleic and arachidonic acids, or through the degradation of 2,4-decadienal [24]. Some branched aldehydes, including 3-methyl-butanal, benzaldehyde, and benzeneacetaldehyde, were detected in the most remarkable abundance in PL compared with other breeds (*p* < 0.05). These three compounds potentially result from the Strecker degradation of leucine, tyrosine, and phenylalanine, respectively [25,26]. Most branched aldehydes contribute to the flavor of grilled meat [24]. Benzaldehyde (14.8–32.3 μg/g) was the most abundant aldehyde and provided an almond odor. It has also been detected as the predominant aldehyde in grilled ribeye, strip loin, and top sirloin steaks [19]. 3-Methyl-butanal and benzeneacetaldehyde, which imparted malty and honey notes, were also reported in both dry-aged and wet-aged beef [27]. 

N-, O-, and S-containing compounds were abundant in Chinese yellow cattle and exhibited distinct odor characteristics. PL samples were the most abundant in pyrazines, which are mainly formed by the condensation of carbonyl ammonia compounds generated by the Strecker degradation reaction [28,29]. Pyrazines have been generally considered to contribute a typical nutty and roasted odor to cooked beef [26]. 2-Acetylpyrrole, imparting nutty and bread notes, was found at a higher level in XJ (1.1 μg/g on average) compared with other breeds. It was the reaction product of dicarbonyl compounds and ammonia [30]. Significantly higher concentrations of methional and 2-acetyl-2-thiazoline were found in JX and XJ than in others (*p* < 0.05). These two compounds played important roles in roast beef [31]. Thiazoles, with the cooking meat or cereal odor, originate mainly from the degradation of thiamine or the Maillard reaction of sulfur-containing amino acids with reducing sugars [32]. The content of 2-pentylfuran (beany, grassy odor) was the highest in PL samples, and it was generated from the oxidation of 2,4-decadienal [33]. 2-Pentylfuran was more abundant in cereal-fed beef than grass-fed beef [24]. Two furanones, namely butyrolactone and octalactone, were also detected with very low levels (0.018, 0.019 μg/g on average) among breeds. Furanones were found to be the dominant volatile substance in Japanese Wagyu cattle [3]. Although N-containing heterocyclic compounds, S-containing compounds, and O-containing heterocyclic compounds were detected at low levels, they were characteristic compounds in meat flavor due to their low odor thresholds [34]. The presence of these components may endow JX, XJ, and PL samples with outstanding performance in flavor.

Most alcohols in WS were significantly higher than those in other breeds (*p* < 0.05). Alcohols have high odor thresholds and are produced from lipid oxidation or carbonyl compound reduction [35]. 1-Octene-3-ol was the most abundant alcohol (0.5–3.2 μg/g). It was the degradation product of linoleic acid and was known as “mushroom alcohol” because of its typical mushroom flavor [36]. 1-Octene-3-ol was found to be the key aroma component of cooked beef meatballs [37]. For the ketones, 3-hydroxy-2-butanone (acetoin) was identified with the highest contents (1.9–6.4 μg/g). The contents of acetoin in XJ and PL were significantly higher than those in other breeds (*p* < 0.05). Acetoin was considered an important contributor to the buttery odor in beef [38], and it is known as the degradation product of dicarbonyl and hydroxyl carbonyl in the Maillard reaction [32]. A diverse array of hydrocarbons was also detected in our study. The concentration of alkylbenzenes (toluene, ethylbenzene, and xylene) was extremely high. In particular, m-xylene was identified at a content of 3.7–26.6 μg/g. Other hydrocarbons included cyclohexanes, aliphatic hydrocarbons, and a small number of terpenes, such as limonene. However, due to the high odor thresholds of hydrocarbons, their effects on beef flavor were weak. Compared to samples from other breeds, there were more low-molecular hydrocarbons in PL and high-molecular hydrocarbons in WS, which may be due to differences in the composition of fatty acids.

### 2.2. Discrimination of Pan-Fried Steaks by PCA, HCA, and OPLS-DA Analysis

PCA is an unsupervised classification method to observe the distribution of samples. The pan-fried steaks from different breeds were discriminated by PCA. As shown in Figure 2a, PC1 and PC2 accounted for 41.4% and 29.0% of the variance, respectively. Moving top to bottom along the Y axis (PC2), BH, XJ, JX, and PL were on the negative side of PC2, while WS and YL were on the positive side. This suggests that the VOCs in BH, XJ, JX, and PL were relatively similar but distinctly different from WS and YL. HCA can be used to depict the similarities and differences among different samples. HCA based on Pearson distance measure and the Ward clustering algorithm was furtherly carried out in our study (Figure 2b). The distance between the clusters shows the degree of similarity between samples. The bigger the distance between two clusters, the more significant the difference in terms of the volatile profiles. At first, YL and WS had the lowest index, indicating that the two breeds have the most similar volatile profiles. At the next stage of the hierarchy, PL and JX clustered together and possessed similar volatiles. Subsequently, PL and JX clustered with XJ and BH in turn. Finally, the six breeds were obviously clustered into two groups. YL and WS have the most similar volatile profiles and were named Group 1. BH, JX, XJ, and PL were named Group 2. The result illustrated that the volatile profile of Group 2 was greatly different from that of Group 1.

According to the classification results of PCA and HCA, supervised OPLS-DA was then performed to verify the validity of clustering and find the key markers to discriminate Group 1 and Group 2. The OPLS-DA score plot (Figure 3a) showed that the two groups could be distinguished clearly (R^2^ = 0.989) and predicted efficiently (Q^2^ = 0.966). The OPLS-DA model was validated by cross-validation ANOVA (*p* < 0.05), indicating the model fitted well. In addition, the importance of various substances was evaluated by variable importance prediction (VIP). VOCs with a VIP score of >1 and *p* < 0.05 were considered as potential markers for the discrimination of different groups [39]. As shown in Figure 3b, 18 VOCs were screened out, including m-xylene, nonanal, hexanal, benzaldehyde, nonanoic acid, toluene, p-xylene, o-xylene, 1-octene-3-ol, 2,5-dimethyl-pyrazine, acetoin, trimethyl-pyrazine, ethylbenzene, benzeneacetaldehyde, 1-octanol, heptanal, octanal, and 2,5-octanedione. Afterwards, S-plot modeling was performed to confirm the potential markers of Group 1 and Group 2 (Figure 3c). The points farther from the origin of the axes represent VOCs that differ more significantly between the two groups. Group 1 comprised mainly aliphatic compounds, such as linear saturated aldehydes, alcohols, and carboxylic acid. Most substances in Group 2 were cyclic compounds, including alkyl benzene, pyrazines, benzaldehyde, and benzeneacetaldehyde. Linear saturated aldehydes, alkylbenzenes, carboxylic acids, and alcohols, which came from the degradation of lipids, made up the main part of the selected 18 VOCs. This suggests that the VOCs differences among the two groups might be attributed to the difference in intramuscular fat (IMF) content or fatty acid composition. Legako et al. (2015) reported that differences in intramuscular lipid content altered the proportion of fatty acids and caused the change in flavor [40]. Frank et al. (2016) compared the grilled beef volatiles from Wagyu and Angus breeds in Australia, and the result showed that the average concentration of most volatiles was higher in Wagyu than in Angus. Differences in the volatiles levels of grilled beef from the two breeds can be explained by the IMF levels, which were positively correlated with a number of volatiles, including key Strecker-derived aldehydes and most of the pyrazines [4]. Gorraiz et al. (2002) evaluated the breed effects on cooked beef flavor from Pirenaica and Friesian. The result that Friesian showed stronger fatty flavor and aftertaste than Pirenaica could be related to their different fatty acid compositions, primarily caused by genetic control of animal lipid metabolism [5]. Research has shown that the indigenous yellow cattle breeds of China distributed in different geographical locations had diverse genetic differentiation [41]. We inferred that yellow cattle breeds with different genetic structures might have different fat deposition capacities, which led to differences in volatile profiles.

### 2.3. Volatile Profiling of Pan-Fried Steaks Obtained by E-Nose

E-nose was also used to analyze the flavor profiles based on the sensitivity ranges of the sensors. The radar chart shows how the mean responses of the sensors differed with the samples (Figure 4a). BH had higher response values to W1W (sensitive to sulfur organic compounds and terpenes) and W2W (sensitive to aromatic compounds and sulfur organic compounds) than other breeds. WS had the highest response to sensor W5S (sensitive to nitrogen oxide) among breeds. The response values of PL to W1C (sensitive to aromatic compounds) and W3S (sensitive to methane) were higher than others. In general, the mean responses of the W1W, W2W, and W5S sensors varied with a wide range, suggesting that N-, S-, and O-containing compounds were different among breeds. 

Furthermore, PCA was performed to investigate the differences between the six breeds. As shown in the PCA score plot (Figure 4b), PC1 and PC2 accounted for 65.8% and 15.6% of the variance, respectively. Although the sample points were partly overlapped, the trend of separation among breeds was observable. The data points from JX, XJ, and PL were mostly located on the positive side of PC1, indicating that the volatiles of these breeds were similar. However, the samples from BH, YL, and WS were on the negative side of PC1, where BH was distributed in the second quadrant, and YL and WS were in the third quadrant. This indicated that the volatile profiles of BH, YL, and WS were similar, and YL and WS had more similar volatiles than BH. 

Both GC-Q-Orbitrap and E-nose, in combination with chemometrics, were effective methods to discriminate the volatile profiles of pan-fried steaks from different Chinese yellow cattle breeds. However, the results were not exactly the same. This may be attributed to the comprehensive and fuzzy analysis of E-nose, resulting in limited access to information, or to the low sensitivity of the E-nose sensors to certain substances, leading to the omittance of these substances. Li et al. (2018) explored the correspondence between the responses of E-nose sensors and VOCs measured by GC-MS, and the result showed that only a portion of volatiles could be captured by the sensors [42]. Bai et al. (2021) also studied the correlation between highly abundant volatiles and E-nose signals and revealed that W1S, W1W, W2S, W2W, and W3S sensors were positively correlated with 6 out of 36 volatiles [43].

### 2.4. PLSR Analysis between Beef Aroma Intensities and Odor-Active Volatiles

The scores of beef aroma intensities are shown in Figure 5a. Aroma scores of PL, XJ, and JX samples were significantly higher than those of the WS and YL samples (*p* < 0.05). The result of aroma scores was the same as that of PCA based on GC-Q-Orbitrap determination. The categories and contents of VOCs had a great effect on the intensity of flavor that people perceived. The contribution of VOCs to flavor was not only related to the content but also to their odor thresholds and interactions, as only specific VOCs with aroma activities could be perceived by people. These odor-active volatiles are the ones that contribute to flavor. To figure out odor-active volatiles (OAVs), the ratio of VOCs concentration to the threshold were calculated. The odor-active volatiles were defined as the VOCs with OAVs ≥ 1 [44]. As shown in Appendix A, 34 odor-active volatiles were found, and they were the main flavor compounds of pan-fried steaks.

To further estimate the contribution of odor-active volatiles to the flavor of Chinese yellow cattle comprehensively, PLSR analysis between the contents of odor-active volatiles measured by GC-Q-Orbitrap and aroma scores was performed. As shown in Figure 5b, the regression model explained 93.5% of the variation in aroma scores. Q^2^ was up to 87.2%, and the permutation test showed that the intercept of the regression line through Q^2^ was negative, indicating that the model fitted well. Figure 5c describes the linear relationship between the contents of odor-active volatiles and the aroma scores. Sample points were distributed on both sides of the regression line. The coefficient of determination, R^2^, was greater than 0.7, indicating good linearity. Figure 5d shows the contribution of odor-active volatiles to the flavor attributes by the regression coefficient. Bars with 95% confidence interval excluding 0 were the significant indications (*p* < 0.05). 3-Methyl-butanal (malty odor), benzeneacetaldehyde (honey/sweet odor), 2-ethyl-6-methyl-pyrazine (roasted odor), 2-acetylpyrrole (caramel/fatty/nutty odor), 2-acetylthiazole (grainy/nutty odor), and 2-acetyl-2-thiazoline (popcorn-like odor) may exert a significant positive influence on the flavor attribute. 

### 2.5. The Difference and Preference Testing for Recombination Samples

The select compounds predictive of aroma intensity were found to be present at lower levels in WS (the lowest score of aroma intensity) compared to PL (the highest score of aroma intensity) (Appendix A). The WS sample was selected as the control group. Highly predictive compounds were added into WS to form a recombination group to match the concentration of PL, and to verify whether they contributed significantly to the beef aroma. The triangle test was used to determine whether panelists perceived differences between the two groups. The results showed that 32 of the 60 panelists chose the one that differed from the other two samples, suggesting that there was a significant difference (α = 0.01) between the control and recombination samples. Therefore, the addition of the six highly predictive compounds induced a perceptible difference in the flavor profile. Meanwhile, preference testing was performed to determine whether the addition of highly predictive compounds was preferred by panelists and had a positive effect on the beef aroma. It was observed that 39 of the 60 panelists (α = 0.05) preferred the recombination group, and they described it as a more intense milky or rice-like aroma. It indicated that the recombination group was more appreciated by panelists compared to the control group. 

The result confirmed that six highly predictive compounds played important roles in Chinese yellow cattle. This was in accordance with the result of MacLeod G et al. [35], who reported that certain carbonyl compounds, sulfides, and pyrroles were probably important contributors to roasted beef. Using olfactometry and aroma extract dilution analysis, Cerny [45] revealed that 2-acetyl-2-thiazoline and 2,3-diethyl-5-methylpyrazine were key compounds that affected the flavor of roasted beef. The research on the flavor characteristics of Angus and Wagyu had shown that the dominant odors of grilled beef were related to 3-methyl-butanal and 2-ethyl-3,5-dimethylpyrazine [4]. Another study on the flavor of Korean cattle also revealed that the contents of 3-methyl-butanal in grilled beef were positively correlated with sensory traits [6]. In addition to the flavor contributors reported previously, we found important roles of benzeneacetaldehyde and 2-acetylpyrrole in Chinese yellow cattle. 

## 3. Materials and Methods

### 3.1. Chemicals

Methanol (HPLC-grade) and n-alkanes standard (C_6_–C_30_) were purchased from Sigma-Aldrich Chemical Co. (Shanghai, China). 2-Methyl-3-heptanone standard was purchased from Dr. Ehrenstorfer GmbH Co. (Augsburg, Germany). 2-Ethyl-6-methyl-pyrazine (97.2%) and 2-acetylpyrrole (99.8%) were obtained from Tan-Mo Technology Co. (Changzhou, China). 3-Methyl-butanal (95%), benzeneacetaldehyde (95%), 2-acetylthiazole (99%), and 2-acetyl-2-thiazoline (97%) were obtained from Aladdin Biochemical Technology Co. (Shanghai, China). Propylene glycol (food-grade) was purchased from Guangzhou Yehu Chemical Co. (Guangzhou, China).

### 3.2. Samples Preparation

A total of 18 strip loins (Choice, according to USDA quality grades) from six Chinese yellow cattle breeds, including Bohai Black Cattle (BH), Jiaxian Red Cattle (JX), Yiling Cattle (YL), Wenshan Cattle (WS), Xinjiang Brown Cattle (XJ), and Pingliang Red Cattle (PL) were used in this experiment. All cattle were grain-fed steers and slaughtered at 36 ± 2.4 months of age with the same commercial slaughter processing. Strip loins of Chinese yellow cattle were vacuum packed and aged at 4 °C for 14 days after slaughter. They were then frozen at −20 °C and transported to the lab at the Institute of Animal Sciences of the Chinese Academy of Agricultural Sciences (CAAS, Beijing, China). The strip loins were immediately sliced into 15 mm thick steaks perpendicular to the direction of the muscle fibers and individually vacuum-packed and stored at −20 °C. After thawing at 4 °C for 2 h, steaks were fried on the pan (200 °C) until their central temperature reached 67 ± 2 °C. Then four slices of each sample were quickly minced, ground with liquid nitrogen, divided into cryopreservation tubes, and stored at −80 °C for GC-Q-Orbitrap, E-nose analysis, and difference and preference testing for recombination. The rested steaks were used for the evaluation of aroma intensity. 

### 3.3. GC-Q-Orbitrap Measurement

Four grams of minced sample were accurately weighed and placed in a sealed glass vial (20 mL). 2-methyl-3-heptanone (0.041 μg/mL, 2 μL) was added as the internal standard. Samples were then equilibrated at 60 °C for 10 min and determined using a GC-Q-Orbitrap system (Q Exactive GC, Thermo Fisher Scientific, Waltham, MA, USA) consisting of a multi-purpose autosampler, a trace 1310 GC, an electron ionization (EI) source, and a Q-Orbitrap mass spectrometer. The solid-phase microextraction with a DVB/CAR/PDMS SPME fiber was used as the autosampler. The fiber was inserted into the headspace of the vial, and the extraction was operated at 60 °C for 30 min. The fiber was then automatically injected into a GC-Q-Orbitrap equipped with a DB-Wax capillary column (30 m × 250 μm × 0.25 μm) (Agilent Technologies, Santa Clara, CA, USA) and was desorbed at 250 °C for 3 min with splitless mode and aged at the same temperature for 20 min. The initial temperature of the GC oven was set at 40 °C for 2 min. The temperature was then raised to 230 °C at a rate of 4 °C/min and maintained for 5 min (run time: 54 min). High-purity helium (99.99% purity) was used as carrier gas and flowed at 1.0 mL/min. Samples were then analyzed with an MS detector in FullScan mode. Electron impact (EI) was used as the ion source, and the ionization energy was 70 eV. The ionization temperature was 280 °C. The quantity scanning range was from 30 to 400. Nitrogen gas (99.99% purity) was used for the C-Trap supply. X-Calibur and Trace Finder 5.1 software (Thermo Fisher Scientific, Waltham, MA, USA) were used for peak identification. Meanwhile, the retention index (RI) was calculated according to the retention times of the n-alkanes standard (C_6_–C_30_). The NIST 2.3 and WILEY 9 libraries were used to identify compounds with a positive and negative similarity index greater than 800. The peak areas of the target compound and the internal standard were used for quantitative analysis.

### 3.4. E-Nose Measurement

E-nose measurement was performed according to the procedure described by Gao et al. (2017) [46], with some modifications. The volatiles profiles of different breeds were discriminated using a PEN3 portable E-nose (Win Muster Airsense Analytics Inc., Airsense, Germany), which was equipped with a sampling apparatus and an array of 10 different metal oxide sensors, named WIC, W3C, W5C, W1S, W2S, W3S, W5S, W6S, W1W, and W2W. One gram of minced sample was placed in a sealed glass vial (10 mL) and equilibrated at room temperature for 30 min, followed by heating at 60 °C for 30 min in a water bath. E-nose was eluted for 5 min, and then the headspace gas was injected into the sensor array chamber via sampling apparatus and detected until the response values of the sensors remained stable. The measurement time was 90 s. An additional 5 min was then required for system rebalance. Each sample was collected three times automatically. 

### 3.5. Sensory Evaluation

#### 3.5.1. Evaluation of Aroma Intensities

Sensory evaluation was conducted in the standard sensory evaluation lab at room temperature (20–23 °C). A panel consisting of 17 panelists (ten females and seven males, age 23–57) assessed the aroma intensity. Panelists were trained according to the procedure of Dikeman et al. (2013) [47] before starting the experiment. The pan-fried steaks without seasoning were cut into portions of 2 cm × 2 cm × 1.5 cm. Four dices of each sample were aliquoted onto plates labeled with a three-digit random code. The samples were stored in an incubator to ensure the temperature remained at 60 °C and presented to the panelists in a randomized order. The aroma intensities were assessed by nose smelling and tasting by the panelists. Panelists were then asked to score the intensity of beef aroma on a typical 9-point scale using the assessment protocol defined during panel training. The lowest score of 1 was given to samples that had an extremely bland beef aroma, and the highest score of 9 indicates an extremely intense beef aroma. All samples (*n* = 18) were evaluated in each test session. Evaluating in triplicate required three sessions. An orientation sample from an extra steak was evaluated, and the scores were given by the panel discussion at one time before each session. Unsalted crackers and drinking water were served to avoid the cross-link effects between samples [48].

#### 3.5.2. The Difference and Preference Testing for Recombination

A total of 599.4 g of minced WS sample spiked with 600 µL of propylene glycol was used as the control group for the difference and preference testing. A stock solution of propylene glycol and highly predictive compounds (total volume of 600 µL) were added to 599.4 g of WS sample, and the mixture was used as the recombination group. The concentrations of highly predictive compounds were determined by the difference between their concentrations in the most intense sample and the WS sample [11]. Four grams of sample were aliquoted into cups with lids and labeled with a three-digit random code.

A triangle test was performed to determine if there was a difference between the control and recombination groups. Sixty panelists (23 females and 37 males, age 26–57) who reported consuming steaks greater than once per month were recruited from the Institute of Animal Sciences of the CAAS. Two samples from the control group and one sample from the recombination group or one sample from the control group and two samples from the recombination group were presented to panelists in a randomized order. Panelists were asked to select which sample was different from the other two samples by nose smelling and tasting. In addition, one sample from the control group and one sample from the recombination group were presented to panelists. Panelists were asked to choose their preferred sample. Testing was also performed in the standard sensory evaluation lab at room temperature (20–23 °C). Unsalted crackers and drinking water were used as a palate cleanser between samples.

### 3.6. Statistical Analysis

One-way analysis of variance (ANOVA) was carried out using SPSS software (IBM SPSS statistics 20.0, IBM Inc., Armonk, NY, USA). The significance of data was compared using Duncan’s multiple range test (*p* < 0.05). Principal component analysis (PCA) and hierarchical cluster analysis (HCA) were performed using MetaboAnalyst 5.0 online. Orthogonal partial least squares-discriminant analysis (OPLS-DA) and PLSR analysis were performed using SIMCA software (SIMCA 14.1, MSK Umetrics Inc., Goettingen, Germany).

## 4. Conclusions, Limitations, and Future Research

Ninety-two VOCs of pan-fried steaks from six Chinese yellow cattle breeds were identified by GC-Q-Orbitrap, and aldehydes were the most abundant components. Six cattle breeds could be divided into two groups by GC-Q-Orbitrap combined with chemometrics. Eighteen VOCs were selected as potential markers for the differentiation of the two groups, and they were mostly the degradation products of lipids. The YL and WS group comprised mainly aliphatic compounds, while most substances in the BH, JX, XJ, and PL group were cyclic compounds. E-nose, in combination with chemometrics, was also an effective method to discriminate the volatile profiles of different breeds, but the results were not exactly the same with GC-Q-Orbitrap due to the different recognition mechanisms for volatiles. PLSR analysis of sensory evaluation and odor-active volatiles identified 3-methylbutanal, benzeneacetaldehyde, 2-ethyl-6-methyl-pyrazine, 2-acetylpyrrole, 2-acetylthiazole, and 2-acetyl-2-thiazoline as key volatiles that affected the flavor attributes of Chinese yellow cattle. Sensory recombination difference and preference testing confirmed that these six highly predictive compounds contributed significantly to the beef aroma. In further study, GC combined with olfactometry technology can be utilized to explore the role of individual flavor component. In addition, further study can be carried out to find the precursors that produce these key volatiles. This can provide directions for nutritional regulation and breeding elite germplasm.

## Figures and Tables

**Figure 1 molecules-27-03593-f001:**
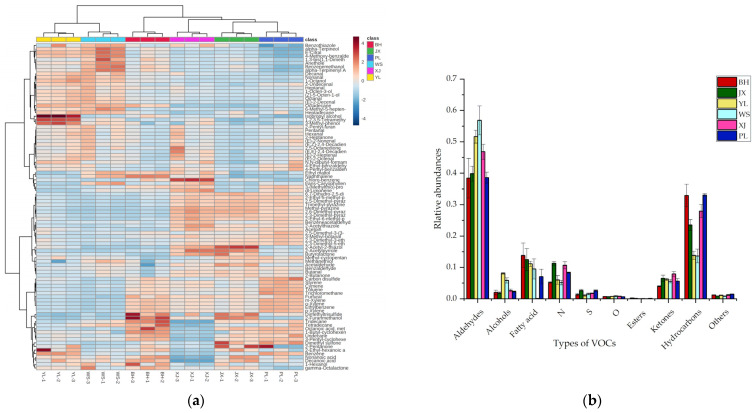
(**a**) Heatmap of VOCs of pan-fried steaks from six different Chinese yellow cattle breeds; (**b**) The relative abundances of different types of VOCs. The abbreviations N, S, O refer to N-containing heterocyclic compounds, S-containing compounds, and O-containing heterocyclic compounds, respectively. BH, Bohai Black Cattle; JX, Jiaxian Red Cattle; YL, Yiling Cattle; WS, Wenshan Cattle; XJ, Xinjiang Brown Cattle; PL, Pingliang Red Cattle.

**Figure 2 molecules-27-03593-f002:**
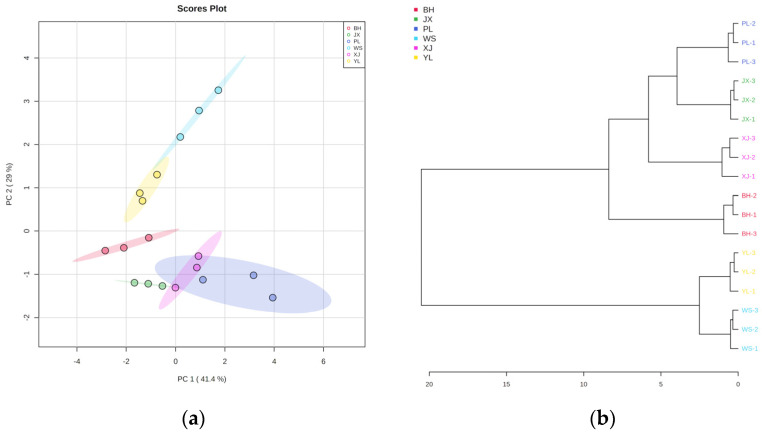
Principal component analysis (PCA) and hierarchical cluster analysis (HCA) of VOCs detected by GC−Q−Orbitrap in six Chinese yellow cattle breeds. (**a**) PCA score plots; (**b**) the dendrogram.

**Figure 3 molecules-27-03593-f003:**
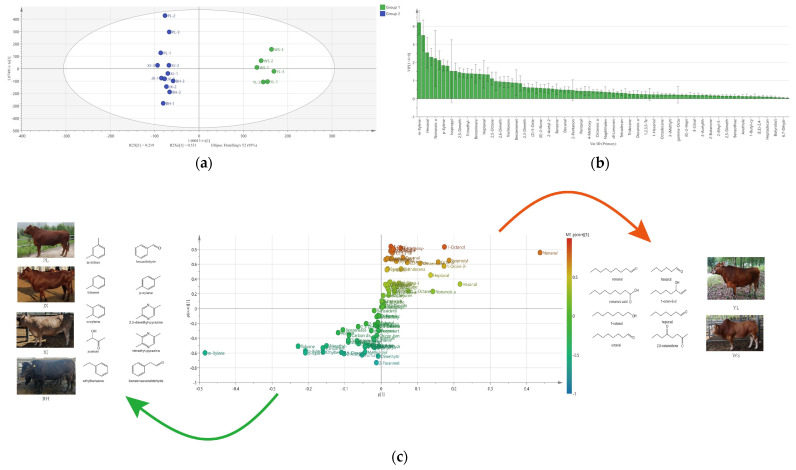
Orthogonal partial least squares-discriminant analysis (OPLS−DA) of VOCs from Group 1 (YL and WS) and Group 2 (BH, JX, XJ, and PL). (**a**) OPLS−DA score plots; (**b**) VIP scores plots; (**c**) S-plot with the chemical structural formula of potential markers of Group 1 and Group 2.

**Figure 4 molecules-27-03593-f004:**
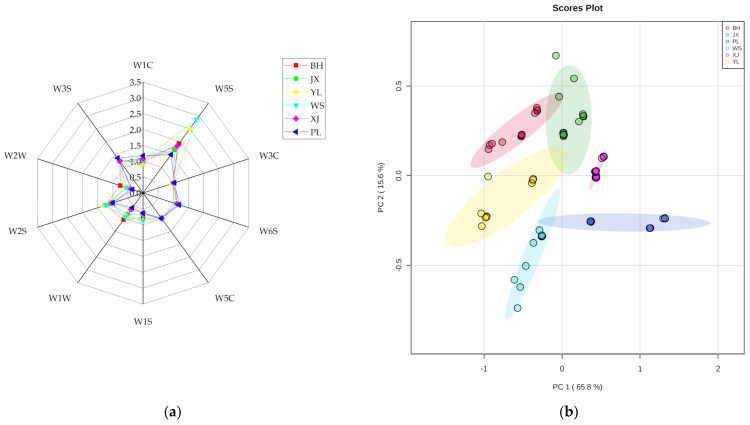
(**a**) Spider diagram and (**b**) PCA score plot of response values detected by E−nose.

**Figure 5 molecules-27-03593-f005:**
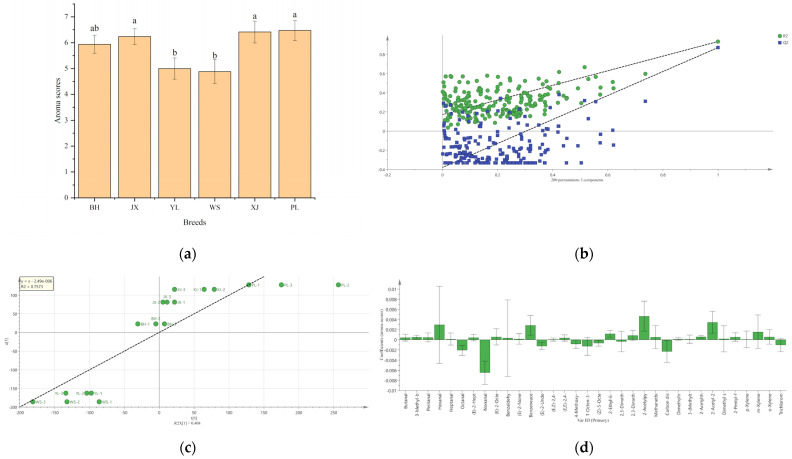
(**a**) Aroma scores evaluated by panel. Mean scores with different letters are significantly different (*p* < 0.05); (**b**–**d**) Partial least square regression (PLSR) analysis for the regression coefficient identification and the significance evaluation of the association between odor-active volatiles and the aroma scores; (**b**) permutation test; (**c**) linear relationship plot; (**d**) the regression coefficient.

## Data Availability

The data presented in this work are available in the article and Appendix A.

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
