# Peer review of "Characterization of Volatile Profiles and Correlated Contributing Compounds in Pan-Fried Steaks from Different Chinese Yellow Cattle Breeds through GC-Q-Orbitrap, E-Nose, and Sensory Evaluation"

_molecules, 2022, doi:10.3390/molecules27113593_

Round 1

Reviewer 1 Report

The paper is very interesting and carefully written. References are well cited and updated. How many comments should be revised before final publication in particularly figures resolution and sized to be clear for readers.  

  • This research should be cited

 https://onlinelibrary.wiley.com/doi/10.1002/fsn3.2019

It uses the same techniques.

  • Size and resolution of figure 1A and 1B should be improved
  • The same should be done for figure 3A,B,C
  • Moreover, figures 5 size and resolution should be improved
  • Chromatogram for table S2 should be displayed. To realize the prominent peaks. Explanation about missed numbers in table S2 should be added under it.

Whereas the authors stated “As shown in Table S2, 34 odor-active volatiles were found, and they were 248 the main flavor compounds of pan-fried steaks. “While numbers are up to 88.  

  • 4. E-nose measurement needs referencing
  • Future research plan and study limitation should be added.

Best wishes

Reviewer 2 Report

The conclusion part (4. Conclusions) should be more explicit and a bit in more detail. The figure size (Figures 1 & 3) is too small to read. They should be sufficiently enlarged to make them readable. If required they may be put in Supporting information. 

The manuscript may be accepted after these corrections.

Reviewer 3 Report

After carefully reading the proposed paper, this paper contains an interesting proposal; my overall impression is that the manuscript presents some results that could be useful in practice. I have a good opinion about this work and recommend its acceptance after addressing the following aspects:

My comments are:

  1. The Abstract is very general. It is necessary to mention a brief description of the content of the manuscript in a clear and concise manner so that the reader can understand the content of the manuscript.
  2. In general, it is usual that section of the introduction presents (in the following order) the topic, motivations of the work, bibliographical review, objectives, the novelty of the manuscript, and description of its sections, with no formulas, which can be moved to a section of background on the topic. This organization must be considered in the revised manuscript.
  3. The quality of all Figures is very poor, this figures must be saved in eps extension, I can not read the data in all figures.
  4. More information around Figures 1 and 2 should be reported.
  5. The authors must provide more details about the computational framework used in the manuscript. For example, software and packages used, features of the computer employed, runtimes, and other computational aspects must be added.
  6. The conclusions need to be improved. Also, the authors must add limitations to the study and more ideas for further research. Then, I suggest titling the final section as "Conclusions, limitations, and future research".

Reviewer 4 Report

the article titled "Characterization of volatile profiles and correlated contributing compounds in pan-fried steaks from different Chinese yellow cattle breeds through GC-Q-Orbitrap, E-nose, and sensory evaluation" is well presented and clearly sets out both the method of preparation of the experimental set-up and the results.
So I propose that it be accepted in its current form

Author Response

Response to Reviewer 4 Comments

Point 1:the article titled "Characterization of volatile profiles and correlated contributing compounds in pan-fried steaks from different Chinese yellow cattle breeds through GC-Q-Orbitrap, E-nose, and sensory evaluation" is well presented and clearly sets out both the method of preparation of the experimental set-up and the results. So I propose that it be accepted in its current form.

Response 1: Thank you for your recognition of our manuscript.

Round 2

Reviewer 1 Report

thanks for your excellent responses